# LncRNA FENDRR Expression Correlates with Tumor Immunogenicity

**DOI:** 10.3390/genes12060897

**Published:** 2021-06-10

**Authors:** Maria Cristina Munteanu, Sri Nandhini Sethuraman, Mohit Pratap Singh, Jerry Malayer, Ashish Ranjan

**Affiliations:** College of Veterinary Medicine, Physiological Sciences, Oklahoma State University, Stillwater, OK 74074, USA; cristina.munteanu@okstate.edu (M.C.M.); Nandhini.sethuraman@okstate.edu (S.N.S.); mohitps@okstate.edu (M.P.S.); jerry.malayer@okstate.edu (J.M.)

**Keywords:** FENDRR, tumor immunogenicity, biomarker, immunotherapy

## Abstract

FENDRR (Fetal-lethal non-coding developmental regulatory RNA, LncRNA FOXF1-AS1) is a recently identified tumor suppressor long non-coding (LncRNA) RNA, and its expression has been linked with epigenetic modulation of the target genes involved in tumor immunity. In this study, we aimed to understand the role of FENDRR in predicting immune-responsiveness and the inflammatory tumor environment. Briefly, FENDRR expression and its relationship to immune activation signals were assessed in murine cell lines. Data suggested that tumor cells (e.g., C26 colon, 4T1 breast) that typically upregulate immune activation genes and the MHC class I molecule exhibited high FENDRR expression levels. Conversely, tumor cells with a generalized downregulation of immune-related gene expression (e.g., B16F10 melanoma) demonstrated low to undetectable FENDRR levels. Mechanistically, the modulation of FENDRR expression enhanced the inflammatory and WNT signaling pathways in tumors. Our early data suggest that FENDRR can play an important role in the development of immune-relevant phenotypes in tumors, and thereby improve cancer immunotherapy.

## 1. Introduction

Long non-coding RNAs (LncRNAs) play an important role in many cellular and developmental processes, including cell proliferation, apoptosis, and organ morphogenesis [1,2]. LncRNA expression is tightly regulated in a cell-specific manner by various intracellular signals [1,3]. Several LncRNAs that regulate chemotherapy resistance and immune modulation in cancer medicine have been reported [4,5]. Of these, FENDRR (Fetal-lethal non-coding developmental regulatory RNA), also known as antisense LncRNA FOXF1-AS1, is a recently identified tumor-suppressor agent. For example, in breast and prostate cancer patients, low FENDRR expression is associated with bone and lymph node metastasis, shorter overall survival, and shorter progression-free survival [6,7]. At the cellular level, low FENDRR expression enhances stem cell-like properties, proliferation, and decreased apoptosis in tumors [8,9,10]. Conversely, higher expression of FENDRR in tumors prevents growth and metastasis in vivo [7,8,10].

A key and unique FENDRR property is its ability to interact with components of Polycomb Repressive Complex 2 (PRC2). PRC2 epigenetically modulates target genes and is an emerging important link in tumor immunity control mechanisms [11,12]. For instance, PRC2-mediated epigenetic modulation of colorectal cancer cells can impact the Th1-type chemokines CXCL9 and CXCL10 to influence the migration of effector T and natural killer cells [13]. FENDRR also augments the expression of the transcription factor RUNX1 to prevent T-regulatory cell expansion [6,14,15]. Additionally, it can competitively bind the microRNA-423-5p, suppressing Treg-mediated immune escape of hepatocellular carcinoma cells [16]. Based on these findings, we hypothesized that FENDRR may be an important molecular determinant of tumor cell immunogenicity.

To test our hypothesis, herein, we characterized the role of FENDRR in melanoma proliferation and nuclear factor kappa β (NF-κB) activation pathways. NF-κB is a transcriptional factor that activates transcription of multiple genes involved in pro-inflammatory responses and apoptotic cell death via cyclin D1 in melanoma cells [17]. We correlated the impact of FENDRR on enhancing NF-κB, intratumoral T cells, and pro-inflammatory cytokine expressions (Interleukin 1 β (IL1β), Tumor necrosis factor α (TNFα), Interferon γ (IFNγ), and transcription factor 1 (TCF1)) in immunogenic murine C26 colon and poorly immunogenic B16 melanoma tumors [17]. B16F10 melanoma cells exhibit reduced expression of MHC class I and co-stimulatory molecules (e.g., CD80, CD86 etc.) and a poor affinity of its self-antigen (TRP-2) to T cell receptors [18,19]. Murine colon cancer is relatively more immunogenic than B16F10s [17]. Reduced secretion of IL1β in a tumor decreases macrophage population and the associated interleukin-10 (IL10) production to enhance the relative proportions of dendritic cells as well as IL2 production [20]. Similarly, robust activation of NF-κB pathway is correlated to the presence of sustained signaling by TNFα, and a defect in IFNγ signaling influences responses to immunotherapy [21]. Further, Tcf1 overexpression in a tumor cell increases the stemness of cancer cells by increasing the accumulations of β-catenin and impacting WNT signaling [22,23]. Our in vitro and in vivo data suggest that FENDRR controls the immunogenicity and proliferation of tumor cells via these pathways to invoke pro-inflammatory and anti-tumor immunity effect.

## 2. Materials and Methods

### 2.1. Cell Culture

Murine B16F10 murine melanoma cells were provided by Dr. Mary Jo Turk at the Geisel School of Medicine at Dartmouth College (Hanover, NH, USA). C26 cell was obtained from the National Cancer Institute. 4T1 breast and LLC lung cancer cells were provided by Dr. Chandan Guha at the Albert Einstein College of Medicine (Bronx, NY, USA). Pan02 cells were obtained from ATCC (Manassas, VA, USA). B16F10 and C26 cells were cultured in Dulbecco Modified Eagle’s medium (DMEM; Corning, Oneonta, NY, USA) containing 10% fetal bovine serum (FBS, Atlanta Biologicals Inc., Atlanta, GA, USA) and 1% streptomycin/penicillin (Gibco, Life Technologies, Waltham, MA, USA). Colon carcinoma (C26) and murine 4T1 breast cancer cell lines, from ATCC, were grown in RPMI 1640 (Corning) containing 10% FBS and 1% streptomycin/penicillin. The ATCC murine pancreatic cell line (Pan02) was grown in Eagle’s Minimum Essential Medium (EMEM, ATCC) containing 10% FBS and 1% streptomycin/penicillin. Human male malignant melanoma tissues, colorectal adenocarcinoma, and normal colon tissues were provided by the Cooperative Human Tissue Network, Southern Division, University of Alabama at Birmingham. Experiments were approved by the Oklahoma State University, Human Use Committee, Institutional Review Board.

### 2.2. In Vivo B16F10 Melanoma and C26 Colon Carcinoma Mouse Models

All animal-related procedures were approved and carried out under the regulations and guidelines of the Oklahoma State University Animal Care and Use Committee. Melanoma was induced in 8–10-week-old C57Bl6 mice by subcutaneous inoculation with 0.5 × 10^6^ B16F10 cells/50 μL phosphate buffered saline (PBS) in the flank region using a 27-gauge needle (BD, Franklin Lakes, NJ, USA). For the C26 colon carcinoma model, 8–10-week old Balb/c female mice (Charles River, Wilmington, MA, USA) were inoculated subcutaneously in the flank region with 0.5 × 10^6^ CT26 cells/50 μL PBS using a 25-gauge needle (BD). Tumor volume was calculated based on serial caliper measurements using the formula (length × width^2^)/2, where length was the largest dimension and width was the smallest dimension perpendicular to length. When the tumors reached a volume of 1500–2000 mm^3^, the mice were sacrificed and the tumor tissues were harvested, snap-frozen in liquid nitrogen, and stored at −80 °C until further analyzed.

### 2.3. Evaluation of Intratumoral Immune Cell Population by Flow Cytometry

Fresh tumor tissue was minced and digested using 200 U/mL collagenase IV (Life Technologies) under constant agitation at 37 °C for 1.5 h. The digested tissue was strained through a 70 μm cell strainer (Corning), and the red blood cells were lysed using RBC lysis buffer (BioLegend, San Diego, CA, USA) according to the manufacturer’s instructions. The cell suspension was stained using 1× PBS containing 2% FBS (staining buffer) in the presence of fluorochrome-conjugated anti-mouse antibodies (BioLegend) for 1 h in the dark on ice. The antibodies used to assess intratumoral immune cell populations were as follows: CD45+, CD3+, CD4+ (CD4+ T or helper Th cells), CD45+, CD3+, and CD8+ (CD8+ T or cytotoxic Tc cells). Fixable Viability Stain 575V (BD Biosciences) was used to stain cell suspensions to exclude dead cells from analysis as per the manufacturer’s instructions, and as we published previously. For detecting IFN-γ, cells were washed after surface marker staining, fixed and permeabilized with transcription factor buffer set (BD Biosciences, Franklin Lakes, NJ, USA), and incubated with APC Cy7 anti-IFN-γ antibody for 30 min in the dark, on ice. The labeled cells were analyzed using the LSRII analyzer (BD) within 24 h post-staining. UltraComp eBeads (Invitrogen, Carlsbad, CA, USA) were used for compensation controls according to the manufacturer’s instructions. Fluorescence-minus-one samples were used as negative controls. Data were analyzed using FlowJo software v.10.2 (Treestar Inc., Ashland, OR, USA).

### 2.4. LncRNA FENDRR Overexpression in B16F10 Cancer Cells

A pcDNA3.1 FENDRR overexpression plasmid was synthesized by Thermo Fisher Scientific (Waltham, MA, USA). B16F10 cells (1 × 10^5^ cells/well) were plated in 12 well plates 24 h prior to transfection. Cells were transfected using Lipofectamine 2000 reagent (Invitrogen, Life Technologies) according to the manufacturer’s protocol. Following incubation at 37 °C for 72 h, cells were collected in TRIzol for RNA extraction and in lysis buffer (T-PER Thermo Fisher Scientific) containing protease and phosphatase inhibitor cocktail for Western blotting.

### 2.5. Smart Pool Small Interfering RNA (siRNA) LncRNA FENDRR Downregulation in C26 Cancer Cells

C26 cells (1 × 10^6^ cells/well) were transfected with 10 pmol/µL Smart pool siRNA (Dharmacon, Inc., Chicago, IL, USA) in the presence of Lipofectamine RNAiMAX (Thermo Fisher Scientific, Invitrogen) according to the manufacturer’s instructions. Cells were collected in TRizol for RNA extraction and relative gene expression analysis.

### 2.6. Western Blotting Analysis of Proliferating Cell Nuclear Antigen (PCNA), Canonical NF-κB, and WNT Signaling Pathway

B16F10 and C26 cells and tumor tissues were lysed in tissue lysis buffer (T-PER) containing a protease and phosphatase inhibitor cocktail (Thermo Fisher Scientific) for 30 min on ice. Cell debris was removed by centrifugation at 15,294× *g* rcf (12,000 rpm) for 15 min at 4 °C. Protein concentration in the cell lysate was determined using a BCA protein assay kit (Pierce, Thermo Fisher Scientific). The proteins (10–20 µg) were separated by 10% SDS-PAGE and subsequently transferred onto a nitrocellulose membrane using the BioRad Turbo Trans system (Hercules, California, CA, USA). After blocking with 5% skimmed milk for 1 h in TTBS (20 mM Tris, 150 mM NaCl, and 0.05% Tween 20, pH 7.5), membranes were incubated overnight with 1:1000 diluted primary antibodies anti-PCNA (Abcam, Cambridge, MA, USA); anti-NF-κB and phospho NF-κB p65, anti-β-catenin, and anti-transcription factor T cell factor 1 (TCF1) (Cell Signaling, Danvers, MA, USA), and anti-Gapdh (Ambion, Thermo Fisher Scientific); as well as anti-mouse β-actin at 1:3000 dilution (Thermo Fisher Scientific). They were then incubated for 1 h with the respective secondary antibody (horseradish peroxidase-conjugated goat anti-mouse or goat anti-rabbit; Jackson Immuno Research, West Grove, PA, USA). Blots were developed using Super Signal West Pico (Thermo Fisher Scientific), and signals were detected with an Amersham Imager 600. Intensity of the bands was quantified by ImageJ densitometry, with β-actin as the loading control.

### 2.7. Quantitative Real-Time Polymerase Chain Reaction (qRT-PCR) for Evaluation of Cytokine Expression in Cancer Cells and Tumor Tissues

RNA from C26 and B16F10 cell lines and tumor tissues was extracted using TRIzol reagent (Ambion, Life Technologies) according to the manufacturer’s instructions. Five µg of total RNA was treated with DNase I (Thermo Fisher Scientific) following the manufacturer’s protocol, followed by phenol chloroform RNA purification. DNase I-treated RNA and 200 U/µL Moloney Murine Leukemia Virus Reverse Transcriptase—MMLV (Invitrogen, Thermo Fisher Scientific)—were used to synthesize cDNA. The RT-PCR reaction was performed with five times-diluted cDNA and specific primers (Appendix A, Table A1) using qPCR Master Mix Plus for SYBR green (Eurogentec, AnaSpec, Fremont, CA, USA) on an Applied Biosystems 7500 fast Real Time PCR instrument. Relative gene expression of LncRNA and mRNA was analyzed by the 2^−ΔCT^ and 2^−ΔΔCT^ method using *GAPDH* as the reference gene.

### 2.8. Evaluation of Pro-Inflammatory Cytokines by ELISA

Interleukin 1 β (IL1β) and tumor necrosis factor α (TNFα) protein levels were measured in the whole tumor lysate by enzyme linked immunosorbent assay (Quantikine ELISA; R&D Systems, Minneapolis, MN, USA) according to the manufacturer’s instructions. IL1β and TNFα protein levels were normalized to the total protein concentration determined using the Pierce BCA protein assay kit.

### 2.9. Statistical Analysis

All experiments were repeated three times. Data are shown as the mean ± standard deviation (SD). One-way analysis of variance followed by Tukey’s post-hoc test was performed for multiple group comparisons, and *p* < 0.05 was considered to be statistically significant. The unpaired T-test was used for analysis of differences between two normally distributed groups using GraphPad Prism 6 software (San Diego, CA, USA).

## 3. Results

### 3.1. FENDRR Expression Correlates with Cancer Cell Immunogenicity

We characterized the FENDRR expression in murine tumor cell lines differing in baseline immunogenicity. Since *β-Actin* and *GAPDH* housekeeping genes did not significantly change the pattern of the target gene expressions, *GAPDH* was utilized as reference gene for data analysis. We found that immunogenic murine tumor cells (e.g., C26 colon, 4T1 breast) exhibited high FENDRR expression levels, whereas moderate to low immunogenic cells (e.g., B16F10 melanoma, Pan02 pancreatic adenocarcinoma, and LLC lung carcinoma) exhibited low to undetectable FENDRR levels (Figure 1A). This relationship also manifested in human patient malignant tumor samples, as higher FENDRR expression levels in colon tumors relative to melanoma was noted (Figure 1B). To understand how FENDRR expression influences the immune phenotype, we knocked down FENDRR expression in C26 cells by 50% (Figure 1C). The reduced expression of FENDRR significantly decreased TNFα, and CXCL10 mRNA and protein expression (Figure 1C–E), suggesting a potential role of FENDRR in tumor cell immunity. Unlike *TNFα*, and *CXCL10*, reduced FENDRR expression in C26 did not significantly decrease *IL1β* gene expression (Figure 1C).

### 3.2. Enhanced FENDRR Expression Is Associated with Immune Cell Infiltration and Pro-Inflammatory Cytokine Expression in Tumors

We monitored mice tumor growth rates for 10–14 d and found that the more immunogenic C26 colon tumor demonstrated relatively slower growth rates than B16F10 melanoma (Figure 2A). Evaluation of FENDRR in the harvested tumors showed ~250-fold higher expression of FENDRR in slow-growing C26 tumors compared to B16F10 melanoma (Figure 2B). The enhanced FENDRR expression in C26 was also associated with a ~50% decrease in *PCNA* gene and protein expression versus that found in B16F10 tumors (Figure 2C–E). Similarly, proliferation marker *Ki67* gene expression was downregulated with increased FENDRR expression in C26 tumors (Figure 2F). To determine whether the tumor growth rates were correlated with the presence of T cell populations, we assessed the immune infiltration rates in the tumor. B16F10 tumors showed marked absence of tumor infiltrating T cells. In contrast, C26 tumors exhibited a slightly enhanced population of CD3+ CD4+ (~12%) and CD3+ CD8+ T cell populations (~4%) associated with increased expression of IFN-γ (Figure 2G,I). The slight increase in CD3+ CD8+ T cell (4%) population in C26 tumors was not statistically significant compared to B16F10. However, the gene expression of pro-inflammatory cytokines was markedly higher in C26 than in B16F10 tumors (~9-, 165-, and 48-fold greater *TNFα*, *ILl1β*, and interferon γ (*IFNγ*) gene expression levels, respectively; Figure 2H), and it mirrored their protein expression profiles, with ~2–3-fold higher levels of TNFα and IL1β and ~50% IFNγ, in C26 relative to B16F10 tumors (Figure 2I,J).

### 3.3. FENDRR Overexpression Reverses Proliferation Rates to Inhibit B16F10 Melanoma Growth Rates

B16F10 cells transfected with a full-length DNA encoding murine FENDRR enhanced its expression by ~5 × 10^6^-fold in B16F10 cells versus untransfected controls (Figure 3A). This reduced the viability of melanoma cells by ~50% (Figure 3B,C) and downregulated PCNAs and Ki67 by ~40 respective ~80-fold in the melanoma cells (Figure 3D–F). In contrast, Lipofectamine 2000 alone and transfections with scrambled plasmids did not induce FENDRR expressions and cell death (not shown). Analysis of pro-inflammatory cytokines (*TNFα*, *IL1β*, and *CXCL10*) expression showed an increase of 15-, 200-, and 30-fold, respectively, compared to the controls (Figure 3G). The pro-inflammatory signaling was associated with increased NF-κB activation by p-65 phosphorylation by ~2–3-fold in transfected B16F10 versus control cells (Figure 3H,I) and a concomitant decrease in the metastatic Wnt signaling markers (β-catenin and TCF1) by 70-fold (Figure 3J–L). Together, FENDDR overexpression inhibited proliferation and induced a pro-inflammatory/immune-stimulatory pathway in melanoma cells.

## 4. Discussion

LncRNAs are a large (>200 nucleotides) and heterogeneous group of non-coding RNAs that are found in the nucleus and cytoplasm [3]. The objective of this study was to evaluate the role of FENDRR LncRNA in tumor cell immunity. We found that FENDRR expression is an important molecular determinant, as its expression correlated strongly with tumor immunogenicity. For example, immunogenic murine tumors (e.g., C26 colon, 4T1 breast) that typically upregulate immune activation genes and MHC class I molecule expression on tumor cells exhibited high FENDRR expression levels (Figure 1A), whereas poorly immunogenic tumor cells (e.g., B16F10 melanoma and LLC lung carcinoma) that show a generalized downregulation of immune-related gene expression exhibited low to undetectable FENDRR levels. We are unaware of any other tumor biomarker yet discovered that correlates with baseline immunogenicity, and we propose that FENDRR should be investigated further to determine its ability to serve such a role.

To verify the impact of FENDRR expression on tumor phenotype, we knocked down its expression rate in C26 colon cancer cells in vitro. We found that FENDRR knock down enhanced the proliferation and reduced the inflammatory profile of cancer cells. In contrast, enhanced FENDRR expression expanded CD4+ and CD8+ T cells and caused higher production of pro-inflammatory cytokines in C26 tumors compared to B16F10 melanoma tumors in vivo. More detailed studies of how FENDRR induces molecular and epigenetic modifications to improve tumor-specific T cells would provide mechanistic insights.

We evaluated the expression of PCNA and KI67 in vitro and in vivo. These are important mediators involved in nucleic acid metabolism and rapid DNA synthesis by tethering the polymerase catalytic unit to the DNA template [24]. Human trials have shown that PCNA can be predictive of survival in patients with malignant melanoma [21]. Our results showed that PCNA markers were downregulated at the gene and protein expression levels, suggesting that FENDRR induces cytotoxic effects via this pathway (Figure 2C–F and Figure 3D–F). We also assessed the role of FENDRR overexpression on Wnt signaling in B16F10 melanoma cells because it is highly expressed in malignant melanoma, drives metastasis by regulating mitochondrial activity in a PTEN-dependent manner, and enhances the formation of vasculogenic mimicry and linearly patterned programmed cell necrosis [22,23,25]. FENDRR-expressing melanoma cells downregulated the Wnt signaling markers β-catenin and TCF1 protein expression by 70-fold (Figure 3I–K). Taken together, these results show that FENDRR impacted the proliferative and metastatic markers to reduce aggressiveness of melanoma tumors.

## 5. Conclusions

A robust pro-inflammatory phenotype is associated with increased FENDRR expression in vivo and in vitro. FENDRR expression correlates to pro-inflammatory cytokine expression and T cell infiltration in tumors, upregulation of the NF-κB pathway, and reduced WNT signaling and cell viability in B16 melanoma cells. Future studies should evaluate the mechanisms involved in these effects to create a translatable pharmacologic approach in clinically relevant models.

## Figures and Tables

**Figure 1 genes-12-00897-f001:**
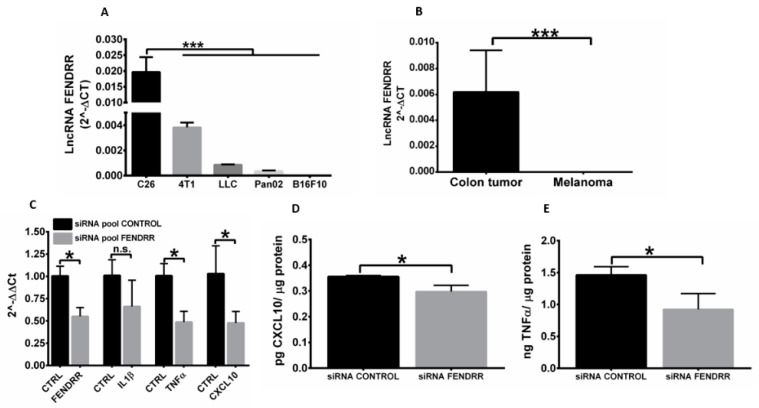
LncRNA FENDRR expression correlates with cancer cell immunogenicity. (**A**) *FENDRR* expression in murine colon (C26), breast (4T1), lung (LLC), pancreatic (Pan02), and melanoma (B16F10) cancer cells shows direct correlation with their baseline immunogenicity (higher in colon and lower in melanoma and pancreatic tumor cells); (**B**) Human LncRNA FENDRR gene expression levels in human colorectal and skin melanoma tumors show significant differences in baseline expression; (**C**) C26 cells were transfected with siRNA pool control or siRNA pool FENDRR for 72 h. *FENDRR*, *IL1β*, *TNFα*, and *CXCL10* were measured by qRT-PCR; (**D**,**E**) FENDRR downregulation inhibited CXCL10 and TNFα protein expression in C26 cells. Data are shown as mean ± SD. * *p* < 0.05, *** *p* < 0.001, n.s. not significant.

**Figure 2 genes-12-00897-f002:**
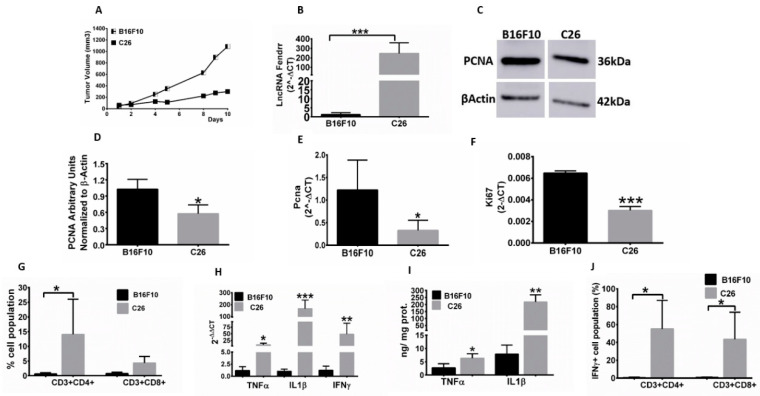
Reduced tumor growth, immune cell infiltration, and tumor pro-inflammatory cytokine expression correlates with enhanced FENDRR expression. (**A**) Slower tumor growth rate in C26 compared to B16F10 tumors (*n* = 5 mice/time point) was noted; (**B**) A 250-fold enhanced *FENDRR* expression in C26 versus B16F10 tumors was noted; (**C**–**E**) PCNA showed 2–3-fold higher protein and gene expression levels in B16F10 versus C26 tumors. PCNA Western blotting densitometry was normalized to β-actin; (**F**) *KI67* showed 2-fold higher gene expression in B16F10 versus C26 tumors. (**G**) Enhanced population of CD4+ and CD8+ T cells in C26 compared to B16F10 tumors; (**H**) Increased expression of *TNFα*, *IL1β*, and *IFNγ* in C26 compared to B16F10 tumors; (**I**) TNFα and IL1β protein expression levels in C26 tumors were significantly higher than those in B16F10 tumors as determined by ELISA; (**J**) CD4+ and CD8+ T cells expressed increased IFNγ in C26 compared to B16F10 tumors; Data are shown as mean ± SD. * *p* < 0.05, ** *p* < 0.01, *** *p* < 0.001.

**Figure 3 genes-12-00897-f003:**
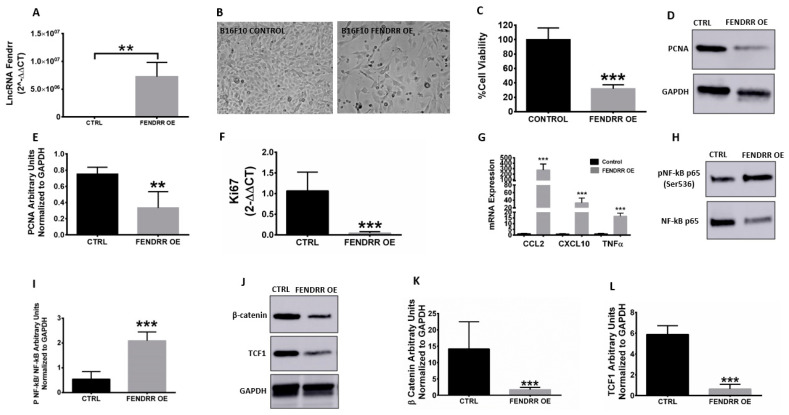
FENDRR overexpression enhances tumor immunogenicity to inhibit B16F10 melanoma growth. (**A**–**C**) Transfection of B16 cells with FENDRR-overexpressing (OE) plasmid significantly increased FENDRR expression and reduced cell viability visually (**B**) and by MTT viability assay (**C**) 72 h after transfection in DMEM serum-free media; (**D**,**E**) PCNA level was significantly decreased in transfected versus untransfected controls (~2–3-fold), as indicated by Western blot analysis using GAPDH as the loading control; (**F**) qRT-PCR analysis showed that *Ki67* gene expression was significantly decreased in transfected versus untransfected controls; (**G**) Immune activating gene expression in FENDRR OE cells was significantly higher than that in the respective controls; (**H**,**I**) Phospho (activated)-NFkB-p65 and total NFkB-p65 protein expression by Western blot, normalized to GAPDH, was significantly elevated by FENDRR OE; (**J**–**L**) Protein expression levels of WNT signaling markers β-Catenin (**J**,**K**) and TCF1 protein expression (**J**–**L**), normalized to GAPDH, were significantly reduced by FENDRR OE. Data are shown as mean ± SD.** *p* < 0.01, *** *p* < 0.001.

## Data Availability

The data supporting the findings of this study are available within the article and from the corresponding author upon request.

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
