# Peer review of "LncRNA FENDRR Expression Correlates with Tumor Immunogenicity"

_genes, 2021, doi:10.3390/genes12060897_

Round 1
Reviewer 1 Report
The main message of the paper is that the FENDRR expression correlates with immunogenicity of studied tumor cells, which could be interesting marker for development of immunotherapies. The data presented in the paper correspond with this conclusion.
However, I see several points where the paper should be improved.
First there are typos omissions and inconsistencies in text and figures:
- The introduction should contain brief info about all cytokines analyzed in the paper as well as WNT pathway.
- As the paper operates with high and low immunogenic cell lines, the source of this classification (citation 19) should be incorporated in the introduction.
- In materials and methods are mentioned human samples and their source. But these samples are not used in the paper. So this parts hould be removed or data from human samples added.
- Cell line names are not consistent throughout the paper - B16F10-B16, C26-CT26, LL/2-LL, LLC. They have to be unified
- On line 70 there is cell line name F16F10 which I believe is typo
- In methods are mentions sources for B16F10 and LLC cell lines but not for the others, so it should be added
- On line 77 -s is stated that "tumors were harvested and stored for further analysis." - What were storage conditions? This should be clearly stated.
- On line 93 - LncRNA FENDRR overexpression in B16F10 and C26 cancer cells - overexpression in C26 cells is not part of the paper
- On line 111 is mentioned that cell debris was removed by centrifugation at 12000 rpm. The value should be presented in rcf
- on line 147 is stated that p<0.5 was considered statistically significant which I believe is a typo.
- On line 137 is stated that 2–ΔΔCT method was used to analyze qPCR data, but in figures are also a lot of plots representing 2–ΔCT. This should be also stated in methods.
- line 159 - we knocked down FENDRR 159 "in C26 cells by 50-fold" - I believe this should be by 50%.
- line 160 - FENDRR decreased IL1β - but it is not significant unlike the rest of measured cytokines and should be stated
- line 184 - CD8+ T population is not significantly larger cd C26 than in B16F10 which should be stated
- Line 222 - LncRNAs are a large (> 80%) and heterogeneous group of non-coding RNAs that are found in the nucleus and cytoplasm - it is not clear fhat what term (>80%) means.
- In figure 1a is not clear what comparisons the p value represents.
- Figure 1d shows blank while figure 1c does not. Blank should be present in both figures or in none of them.
- Figure 2a - tumor volume is missing units and it is hard to distinguish which line represents which cell line.
- Figure 3 lacks caption. It is the same one as Figure 2!
Apart from these issues there are some more general issues:
- In methods it is stated that "The labeled cells were analyzed using the LSRII analyzer (BD) within 24 h post-staining." This is too long for reliable analysis. But to make final conclusion it is critical to add information about the storage of the cells after staining and eventual fixaton if it was present. Also information how was ensured that only viable cells were analyzed is missing and should be added.
- In methods it is stated, that on blots were detected Gapdh and β-actin. But in figures is always shown only one of them. They should be shown both so it could be determined that the change is not effect of loading or change of expression of one housekeeping gene.
- Use of only one housekeepinge gene for qPCR (and especially Gapdh) is insufficient. For reliable results authors should add at least two more housekeeping genes to analysis.
- From chapter 2.4 of Materials and Methods it is apparent that in the overexpression study are compared cells transfected with FENDRR plasmid with cells that were not transfected at all. Thus it is not possible to discern what of the effects on cell viability and gene expression ascribed to FENDRR overexpression are a result of overexpression and which are effects of Lipofectamine 2000 transfection reagent cytotoxicity. Data from mock transfected cells has to be added, otherwise FENDRR effects on cell viability and cytokines expression cannot be concluded.
Author Response
The main message of the paper is that the FENDRR expression correlates with immunogenicity of studied tumor cells, which could be interesting marker for development of immunotherapies. The data presented in the paper correspond with this conclusion.
However, I see several points where the paper should be improved.
First there are typos omissions and inconsistencies in text and figures:
- The introduction should contain brief info about all cytokines analyzed in the paper as well as WNT pathway.
RESPONSE 1: The introduction was revised to include a brief info on the inflammatory pathways (line 46-59)
- As the paper operates with high and low immunogenic cell lines, the source of this classification (citation 19) should be incorporated in the introduction.
RESPONSE 2: We modified the introduction to include this revision (line 55).
- In materials and methods are mentioned human samples and their source. But these samples are not used in the paper. So this part should be removed or data from human samples added.
RESPONSE 3: The results on the human samples are presented in Figure 1B and discussed in the results section, line 155-157.
- Cell line names are not consistent throughout the paper - B16F10-B16, C26-CT26, LL/2-LL, LLC. They have to be unified
RESPONSE 4: The cell line names were unified throughout the paper, as suggested.
- On line 70 there is cell line name F16F10 which I believe is typo
RESPONSE 5: Line 70 was corrected as suggested
- In methods are mentions sources for B16F10 and LLC cell lines but not for the others, so it should be added line 58-59
RESPONSE 6: The cell line sources for C26, 4T1 and Pan02 was updated in the methods section.
- On line 77 -s is stated that "tumors were harvested and stored for further analysis." - What were storage conditions? This should be clearly stated.
RESPONSE 7: The storage conditions were updated, in the methods section, line 77-78.
- On line 93 - LncRNA FENDRR overexpression in B16F10 and C26 cancer cells - overexpression in C26 cells is not part of the paper
RESPONSE 8: C26 was removed from the title, line 96.
- On line 111 is mentioned that cell debris was removed by centrifugation at 12000 rpm. The value should be presented in rcf
RESPONSE 9: 12,000 x rpm was corrected to 15,294 x rcf, line 117
- on line 147 is stated that p<0.5 was considered statistically significant which I believe is a typo.
RESPONSE 10: Typo corrected, p<0.05 was considered statistically significant, line 155.
- On line 137 is stated that 2–ΔΔCT method was used to analyze qPCR data, but in figures are also a lot of plots representing 2–ΔCT. This should be also stated in methods.
RESPONSE 11: 2-ΔCT has now been stated in the method section, line 141.
- line 159 - we knocked down FENDRR 159 "in C26 cells by 50-fold" - I believe this should be by 50%.
RESPONSE 12: Correction was included in line 165.
- line 160 - FENDRR decreased IL1β - but it is not significant unlike the rest of measured cytokines and should be stated
RESPONSE 13: Additional discussion was added to the results section to described it, line 165-169.
- line 184 - CD8+ T population is not significantly larger cd C26 than in B16F10 which should be stated
RESPONSE 14: This was updated in the results section, line 192-193.
- Line 222 - LncRNAs are a large (> 80%) and heterogeneous group of non-coding RNAs that are found in the nucleus and cytoplasm - it is not clear fhat what term (>80%) means. –
RESPONSE 15: This sentence was modified to read as LncRNAs are a large (> 200 nucleotides) and heterogeneous group of non-coding RNAs (line 241).
- In figure 1a is not clear what comparisons the p value represents.
RESPONSE 16: Fig.1a, the group comparison was added to the plot.
- Figure 1d shows blank while figure 1c does not. Blank should be present in both figures or in none of them.
RESPONSE 17: Fig. 1d has been updated and the blank samples were removed from the plot.
- Figure 2a - tumor volume is missing units and it is hard to distinguish which line represents which cell line.
RESPONSE 18: Figure 2a was updated with the missing units.
- Figure 3 lacks caption. It is the same one as Figure 2
RESPONSE 19: Fig. 3 legend was updated in the revised.
Apart from these issues there are some more general issues:
- In methods it is stated that "The labeled cells were analyzed using the LSRII analyzer (BD) within 24 h post-staining." This is too long for reliable analysis. But to make final conclusion it is critical to add information about the storage of the cells after staining and eventual fixaton if it was present. Also information how was ensured that only viable cells were analyzed is missing and should be added.
RESPONSE 1: Fixable Viability Stain 575V (BD Biosciences) was used to stain cell suspensions to exclude dead cells from analysis as per the manufacturer's instructions in methods (line 102).
- In methods it is stated, that on blots were detected Gapdh and β-actin. But in figures is always shown only one of them. They should be shown both so it could be determined that the change is not effect of loading or change of expression of one housekeeping gene.
RESPONSE 2: β-Actin and Gapdh gene and protein expression level did not differ between various experimental groups, so either of them were utilized for protein normalization. Example: PCNA in tumor tissues was normalized to β-Actin, while GAPDH was used to normalize the protein expression in the B16F10 cells.
- Use of only one housekeepinge gene for qPCR (and especially Gapdh) is insufficient. For reliable results authors should add at least two more housekeeping genes to analysis.
RESPONSE 3: For preliminary gene expression screenings, both β-Actin and Gapdh were used as housekeeping genes. They did not significantly change the pattern of the target genes analyzed in our study. Thus, we decided to use only Gapdh as reference gene for all the gene expression analysis study in the manuscript.
- From chapter 2.4 of Materials and Methods it is apparent that in the overexpression study are compared cells transfected with FENDRR plasmid with cells that were not transfected at all. Thus it is not possible to discern what of the effects on cell viability and gene expression ascribed to FENDRR overexpression are a result of overexpression and which are effects of Lipofectamine 2000 transfection reagent cytotoxicity. Data from mock transfected cells has to be added, otherwise FENDRR effects on cell viability and cytokines expression cannot be concluded.
RESPONSE 4: Lipofectamine 2000 by itself was minimally toxic to cells. We also performed studies with scrambled plasmids, and they didn't induce cell death. A more detailed publication incorporating these controls is currently in works. The intent of this publication is to drive home the potential role of FENDRR in tumor immunogenicity, and its relationships with various tumor types. Data suggests that FENDRR can influence inflammatory and cytotoxic pathways in tumor cells.
Reviewer 2 Report
Munteanu et al have correlated FENDRR expression with immunogenicity. The authors show convincingly that increased FENDRR expression is associated with pro-inflammatory phenotype. Their report is of interest.
My suggestions for improvement are the following:
Results:
Figure 1d. The authors should show the expression levels of IL1beta and TNFALFA proteins (western blot), and their correlation with FENDERR expression.
In the line 180. Is PCNA the best marker of cell growth? Ki67 is another example, which could be added in the results.
Figure 2c. This figure is not clear.
Figure 2h. The authors should show the expression levels of IFN gamma protein (western blot).
Figure 3d. The bands are not normalized.
Author Response
Comments and Suggestions for Authors
Munteanu et al have correlated FENDRR expression with immunogenicity. The authors show convincingly that increased FENDRR expression is associated with pro-inflammatory phenotype. Their report is of interest.
My suggestions for improvement are the following:
Results:
- Figure 1d. The authors should show the expression levels of IL1beta and TNFALFA proteins (western blot), and their correlation with FENDERR expression.
RESPONSE 5: Additional data were included for TNFα protein expression in C26, transfected with siRNA FENDRR (fig. 1e). Gene expression analysis of IL1β, TNFα and CXCL10 was performed, however, IL1β didn't significantly differ between FENDRR downregulated vs. siRNA control cells. Thus, only CXCL10 and TNFα were assessed at the protein expression levels (Fig.1d-e).
- In the line 180. Is PCNA the best marker of cell growth? Ki67 is another example, which could be added in the results.
RESPONSE 6: As suggested, the gene expression of Ki67 in murine melanoma and colon tumors, as well as in B16F10s overexpressing FENDRR vs. untreated control was assessed, and they showed similar patterns as the PCNA expression with FENDRR (Fig. 2f).
- Figure 2c. This figure is not clear.
RESPONSE 7: Fig. 2c is a representative image of our findings. Densitometric analysis was performed in triplicates.
- Figure 2h. The authors should show the expression levels of IFN gamma protein (western blot).
RESPONSE 8: This is a great suggestion. We now include the flow cytometric analysis of IFNg results in the tumor tissues (Fig. 2j). Data suggest that the more immunogenic tumors were relatively richer in IFNg production compared to the immunosuppressive murine melanomas.
- Figure 3d. The bands are not normalized.
RESPONSE 9: The bands were normalized to the GAPDH as shown in Fig. 3e.
Round 2
Reviewer 1 Report
Authors addressed almost all my points.
The remaining ones are responses 3 and 4 below.
RESPONSE 3: For preliminary gene expression screenings, both β-Actin and Gapdh were used as housekeeping genes. They did not significantly change the pattern of the target genes analyzed in our study. Thus, we decided to use only Gapdh as reference gene for all the gene expression analysis study in the manuscript.
RESPONSE 4: Lipofectamine 2000 by itself was minimally toxic to cells. We also performed studies with scrambled plasmids, and they didn't induce cell death. A more detailed publication incorporating these controls is currently in works. The intent of this publication is to drive home the potential role of FENDRR in tumor immunogenicity, and its relationships with various tumor types. Data suggests that FENDRR can influence inflammatory and cytotoxic pathways in tumor cells.
Both underscored informations should be incorporated into the main text, to make readers aware about these facts.
Author Response
RESPONSE 3: For preliminary gene expression screenings, both β-Actin and Gapdh were used as housekeeping genes. They did not significantly change the pattern of the target genes analyzed in our study. Thus, we decided to use only Gapdh as reference gene for all the gene expression analysis study in the manuscript.
RESPONSE: The results section 3.1 was revised to include additional information regarding gene expression analysis approach used in our study (line 176 – 178).
RESPONSE 4: Lipofectamine 2000 by itself was minimally toxic to cells. We also performed studies with scrambled plasmids, and they didn't induce cell death. A more detailed publication incorporating these controls is currently in works. The intent of this publication is to drive home the potential role of FENDRR in tumor immunogenicity, and its relationships with various tumor types. Data suggests that FENDRR can influence inflammatory and cytotoxic pathways in tumor cells.
RESPONSE 4: The results section 3.3 was revised to include additional information on scrambled plasmids and Lipofectamine 2000 cytotoxicity (line 261 – 263).
ADDITIONAL CHANGES:
We also briefly revised our references section as follows:
Additional ref. 7 included (line 30)
Additional ref. 6 included (line 39)
Ref. correction (line 288)
Ref. correction (line 310)
Ref. correction (line 317-318)